# Characteristics of ocular-related emergency department visits: Five-years data from a tertiary care center in Riyadh, Saudi Arabia

Abdulmohsen Alshammari[1,2]*, Naif Alqadhy[1,2], Abdulaziz Gharawi[1,2], Bader Alqahtani[1,2], Saif Alagha[1,2], Mohammed AlShenaiber[1,2], Fahad Almalik[1,2], Abdullah Alshibani[2,3]

1 College of Medicine, King Saud bin Abdulaziz University for Health Sciences, Riyadh, Saudi Arabia, 2 King Abdullah International Medical Research Center, Riyadh, Saudi Arabia, 3 Emergency Medical Services Department, College of Applied Medical Sciences, Riyadh, Saudi Arabia

* Abdolmohsens.2001@gmail.com

**Editor:** Vaitheeswaran Ganesan Lalgudi, University at Buffalo Jacobs School of Medicine and Biomedical Sciences: University at Buffalo School of Medicine and Biomedical Sciences, UNITED STATES OF AMERICA

## Abstract

### Background

Ocular emergencies (OE) include ocular trauma, infections, retinal detachment, and uveitis. Due to the limited number of studies describing ocular emergencies requiring Emergency Department (ED) visits in Saudi Arabia, there is a need for further studies addressing this topic. Therefore, this study aimed to assess and describe the incidence, clinical presentation, and causes of ocular-related ED visits at a tertiary hospital in Riyadh, Saudi Arabia.

### Methods

This was a retrospective cohort study. Medical records were reviewed for all patients (all age groups) patients presenting at the ED from 1 January 2019 to 31 October 2023 where data for patients with any ocular emergency was extracted using an Excel sheet. Data analysis was performed using RStudio (R version 4.3.1). Baseline information was described using frequencies and proportions. Types of Ocular emergencies were described using the International Classification of Diseases, 10th Revision (ICD-10) codes and presented using frequencies and proportions. The distribution of ocular emergencies across months and age groups was expressed using chart figures.

### Results

A total of 15,321 ocular-related ED visits were included in this study. Almost 51% of patients were males. The mean age at diagnosis was 25.5 ± 22.1 years. More than half of the patients (51.0%) were diagnosed during childhood (<18 years), while proportions of older ages declined gradually. Average presentations of ocular emergencies in a single month ranged from 200 to 400. Conjunctival disorders including conjunctivitis (29.8%) were the most frequent ocular emergencies.

**Data Availability Statement:** Data cannot be shared publicly because of the ploicy of the research center providing the ethical approval. Data are available from the King Abdullah International Reseach Center, Instiutional Review Board Committee (contact via KAIMRC@NGHA.MED.SA) for researchers who meet the criteria for access to confidential data.

**Funding:** The author(s) received no specific funding for this work.

**Competing interests:** The authors have declared that no competing interests exist.

## Conclusion

The findings of this study showed that ocular emergencies are more prevalent in younger patients (aged <18 years), more commonly due to conjunctivitis. This highlights the need for policymakers to assess the causes of such emergency in this population and implement prevention strategies. Moreover, an average of 200 to 400 emergency visits per month are ocular-related. This finding could help policymakers understand the burden of ocular emergencies on the ED and the pressure that could add to the ED staff to provide appropriate care for these patients.

## Introduction

Emergency departments (EDs) face an increasing demand for effective clinical care. Ophthalmology has a high volume of emergencies, and a progressive increase in ophthalmological emergencies (OE) has been noted in recent years [1, 2]. Ophthalmic emergencies include ocular trauma, infections, retinal detachment, and uveitis, in which there are sudden risks to the visual system [3]. If not adequately treated, these conditions can result in permanent vision loss or severe threats to the patient's visual function.

A cohort study conducted in the United States of America (USA) showed that over a period of six years, an estimated 11,929,955 visits to the emergency departments accounted for ophthalmic conditions. Additionally, corneal abrasions (13.7%) and foreign body in the external eye (7.5%) were the most common presentations in the emergency department [3]. Recent evidence showed that the most common ophthalmology-specific ED visits were diagnosed as viral conjunctivitis (8.7%), dry eye syndrome (6.6%), and corneal abrasion (6.6%) [4]. A study conducted in China showed that out of 1907 patients, 30.5% were classified as "non-emergency" and 23.5% were classified as "emergency" [5]. The most common complaints were red eye (69.7%), eye pain (53%), ocular trauma (44.1%), tearing (43.6%), and finally blurred vision (43.1%) (5). Furthermore, the leading diagnoses were ocular trauma (44.1%), conjunctival disease (23.5%), and vitreoretinal disease (12.1%) [5]. In Lebanon, a study found that conjunctivitis (31.8%), subconjunctival hemorrhage (27.4%), and keratitis (6.6%) were the most common ocular presentations in the emergency department in 2012 [6]. In Saudi Arabia, a study was conducted to determine the frequency and various ocular-related diagnoses of patients presenting to the emergency department [7]. Of the 868 patients, 282 (32.5%) were reported to have conjunctivitis, making it the most common diagnosis, followed by dry eye (18.0%), and lid infections (12.0%) [7]. Moreover, another study from Saudi Arabia investigated the characteristics of patients attending the emergency department and the pattern of ocular emergency cases [8]. Among 1,412 patients, the most frequent diagnosis was trauma (27%), followed by conjunctivitis (14.9%), lids and lacrimal system problems (9.4%) [8].

The available evidence from Saudi Arabia is limited, as few studies were conducted to describe ocular-related emergencies requiring ED visits. Furthermore, the available evidence included a limited number of patients over a short period of time. This highlights the need for further studies exploring this topic on a larger scale and over a longer period to better describe ocular-related emergency visits in Saudi Arabia. Therefore, this study aimed to describe ocular-related emergency visits at one of the largest EDs in the Middle East at a tertiary hospital in Riyadh, Saudi Arabia over five years from January 1st, 2019, to October 31st, 2023.

## Methods

### Study setting

The study was conducted in the ED at King Abdulaziz Medical City (KAMC), Riyadh, which is considered one of the most comprehensive healthcare medical cities in Saudi Arabia. King Abdulaziz Medical City (KAMC) was established in 1982, and it currently has 1973 operational beds and employs about eight thousand health and medical support professionals. Ethical approval was obtained from King Abdullah International Medical Research Center (KAIMRC) (Institutional Review Board approval No.: IRB/2939/23). As this study was a retrospective study using routine hospital data, all obtained data were fully anonymized before it was accessed. Data was accessed on 23 November 2023. All data was anonymized and kept on a university secured desktop where only the study investigators have access to.

### Study design

This study is a retrospective cohort study based on a chart review of the patients admitted to the emergency department between 2019–2023. It is an observational study, and the primary source of information was patient medical charts using the "BestCare" system, which is an electronic patient medical records system that is employed at KAMC.

### Study population

All patients from all age groups who presented to the ED with ocular-related conditions were included in this study. Patients with missing information age, gender, country, were excluded from the study.

### Data collection and management

Data was collected by the study team members using an EXCEL data collection sheet which included the required information for all patients presenting at the ED in KAMC between 2019–2023 with ocular-related emergency. A pre-determined EXCEL data collection form was used to collect data. The data collection form contained patients' demographics information (age, gender, country, and comorbidities) and ocular-emergency related information (date of ED presentation, main diagnosis based on the internation Classification of Diseases– 10th Edition [ICD-10] codes, and type of management). All collected data were anonymized and kept secured at a university-secured desktop and only the study team has access to this data.

### Data analysis

Data analysis was performed using RStudio (R version 4.3.1). Descriptive analyses were conducted as categorical variables were expressed in frequencies and percentages and continuous variables were expressed in mean and Standard Deviation (SD). A line chart was developed to present monthly frequencies of ocular emergency cases. The most common ages upon diagnosis were visualized in a histogram, and age categories were created and expressed as <18 years, as well as between >18 and 30 years, >30 and 45 years, >45 and 60 years, and over 60 years. The ICD-10 codes were used to classify and present the main diagnoses.

## Results

In the current study, 15,464 were initially included to have ocular-related ED. After excluding patients with missing information (n = 143), a total of 15,321 were finally included in this study (**Fig 1**).

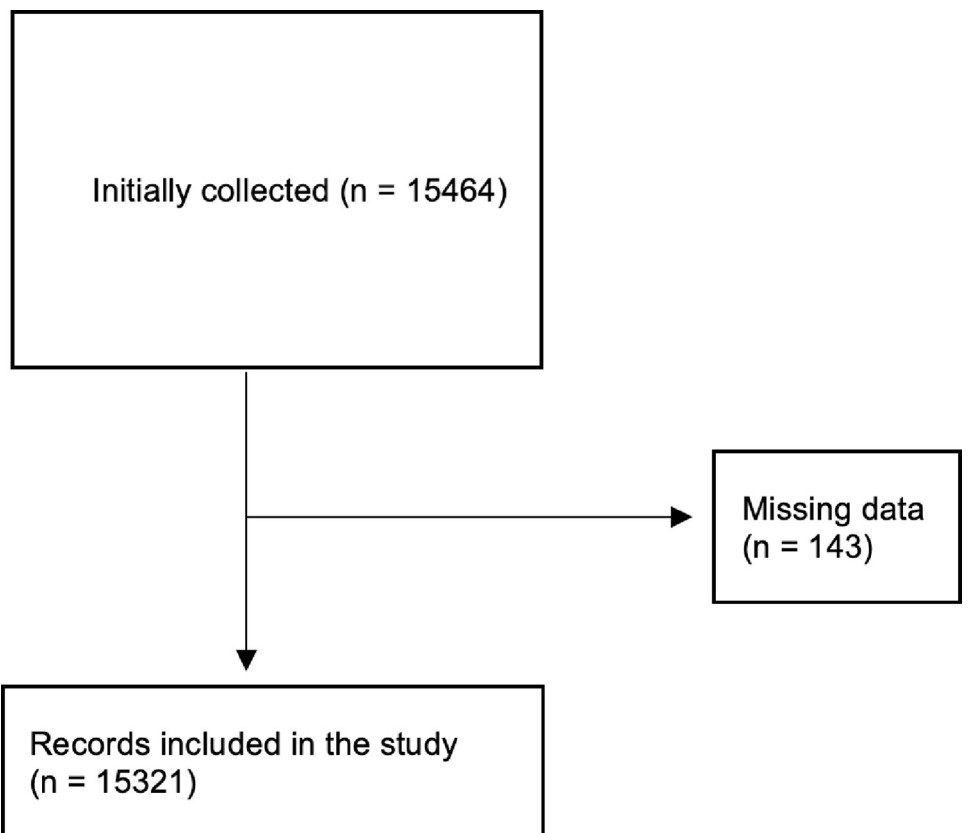

**Fig 1. Flowchart of included patients.**

The mean age of the study population was 25.5 ± 22.1 years, with an almost equal gender distribution (50.7% were males). Of the study population, 1,954 (12.8%) had comorbidities; dyslipidemia was the most prevalent comorbidity (5.8%), followed by hypertension (3.2%) and stroke (1.7%). The majority of patients were from Saudi citizens (96.4%), followed by smaller proportions of patients from Asian countries such as the Philippines (1.2%) and Malaysia (0.5%) (**Table 1**).

The number of ocular-related ED visits by month in 2019 onwards ranged from 250 to 400 cases per month. However, beginning in March 2020, these figures decreased, remaining below 300 cases monthly until March 2021. Subsequently, the number of emergency cases fluctuated between 200 and 300 from April 2021 to October 2023 (**Fig 2**).

In terms of the age distribution at the time of diagnosis for ocular emergencies, findings indicated that approximately half of the patients were diagnosed during childhood (<18 years, n = 7822, 51.0%). The proportion of ocular emergencies gradually declined with increasing age: 16.1% between >18 and 30 years, 12.1% between >30 and 45 years, 10.8% between >45 and 60 years, and 10.2% over 60 years (**Fig 3**).

## Distribution of common ocular emergencies

The most frequent presentation of ocular emergencies at the ED was conjunctival-related disorder, including conjunctivitis (n = 4559, 29.8%) and other disorders of conjunctiva (n = 566, 3.7%). This was followed by injury to the eye and orbit classification, which included injury to the conjunctiva and corneal abrasion without mention of the foreign body (n = 3570, 23.3%)

**Table 1. Characteristics of the study population.**

| Characteristic | N = 15321 |
|---|---|
| **Age upon diagnosis** | 25.5 ± 22.1 |
| **Gender** | |
| Male | 7,774 (50.7%) |
| Female | 7,547 (49.3%) |
| **Comorbidities** | |
| Dyslipidemia | 889 (5.8%) |
| HTN | 486 (3.2%) |
| Stroke | 259 (1.7%) |
| Heart failure | 229 (1.5%) |
| DM | 91 (0.6%) |
| **Country** | |
| Saudi Arabia | 14,771 (96.4%) |
| Philippines | 189 (1.2%) |
| Malaysia | 69 (0.5%) |
| Other Asian Countries | 165 (1.1%) |
| Other African Countries | 98 (0.6%) |
| Other European Countries | 17 (0.1%) |
| United States of America | 8 (0.1%) |
| Australia | 2 (0.0%) |
| Canada | 2 (0.0%) |

Mean ± SD; n (%)

and unspecified eye and orbit injuries (n = 494, 3.2%). The third most common category was other disorders of the eye and adnexa, in which unspecified disorders of the eye and adnexa accounted for most of the diagnoses (n = 1834, 12%). This was followed by disorders of the eyelid, lacrimal system, and orbit, which incorporated hordeolum and chalazion (n = 575, 3.8%) and other inflammation of the eyelid (n = 371, 2.4%) (**Table 2**).

## Discussion

This study has described ocular-related ED visits at a tertiary hospital in Riyadh, Saudi Arabia. More than 15,000 ocular conditions requiring ED visits were included in this study. To our knowledge, this is the largest and most recent representative epidemiological study to describe ocular-related emergencies in Saudi Arabia. Almost 51% of patients were males. The mean age at diagnosis was 25.5 ± 22.1 years. The dominant age group at the time of diagnosis was during childhood (<18 years), being more than half of the patients (51.0%), while proportions of older ages declined gradually. Ranging from 200 to 400, was the average number of presentations of ocular emergencies in a single month. The most frequent ocular emergencies were conjunctival disorders including conjunctivitis (29.8%).

In this study, it was found that the highest number of presentations in a single month did not exceed 400, which is much lower compared to Alotaibi et al. [8] single-month study (1,412 patients with a daily average of 47 patients), though it was conducted in the same city, Riyadh city. This could be attributed to the fact that the study setting in Alotaibi et al. [8] study is a tertiary hospital that is specialized in ophthalmology and Ear, Nose, and Throat (ENT); potentially leading to more patients living in Riyadh to visit a specialized hospital for their ophthalmology/ENT related complaints. The location of the study setting could also play an

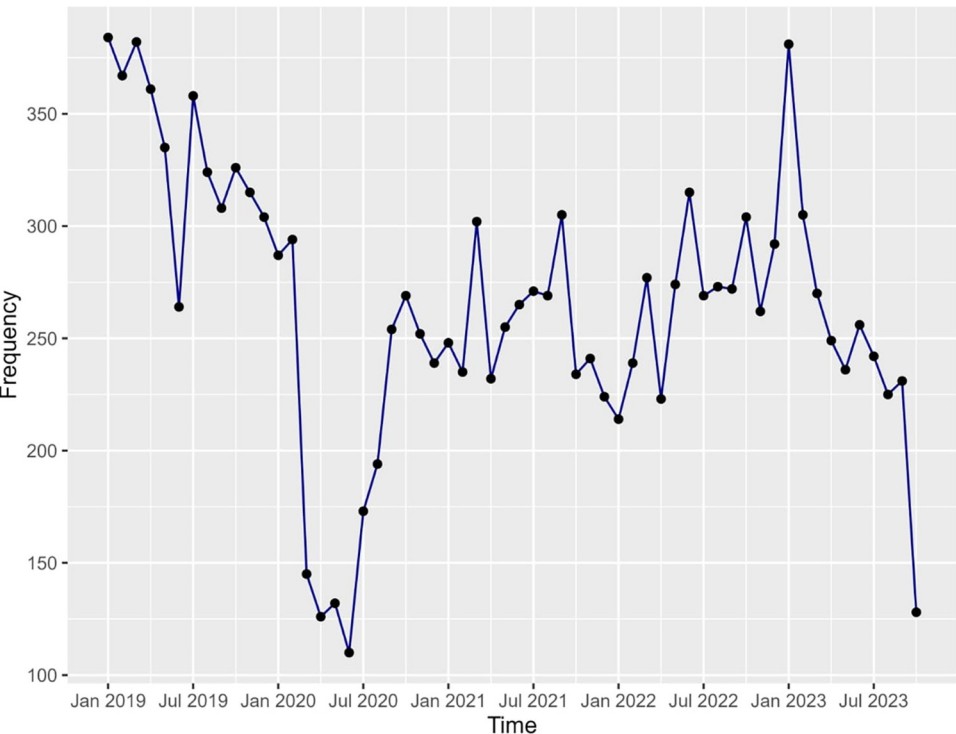

**Fig 2. The number of ocular-related emergency department visits by month.**

important role, as the study setting in Alotaibi et al. [8] study is King Abdulaziz University Hospital, which is located closer to the center of the city of Riyadh compared to KAMC, making it easier to access by more patients especially in the case of an emergency. The average

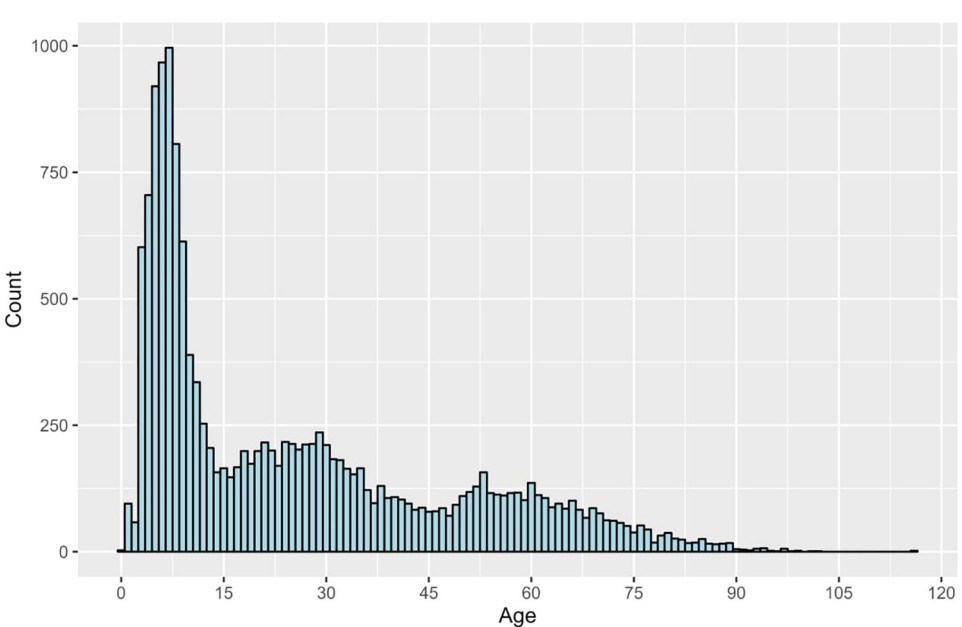

**Fig 3. The frequency distribution of age upon diagnosis of ocular emergencies.**

**Table 2. Distribution of conditions attending the emergency department at King Abdulaziz Medical City (KAMC) in Riyadh, Saudi Arabia.**

| Characteristic | N = 15,321 |
|---|---|
| **Category and diagnosis name** | |
| Disorders of conjunctiva | 5129 (33.5%) |
| Injury of eye and orbit | 4182 (27.3%) |
| Other disorders of eye and adnexa | 1,859 (12.1%) |
| Disorders of eyelid, lacrimal system and orbit | 1,473 (9.6%) |
| Disorders of vitreous body and globe | 612 (4.0%) |
| Visual disturbances and blindness | 576 (3.8%) |
| Disorders of sclera, cornea, iris and ciliary body | 543 (3.5%) |
| Disorders of ocular muscles, binocular movement, accommodation and refraction | 240 (1.6%) |
| Disorders of lens | 206 (1.4%) |
| Glaucoma | 176 (1.2%) |
| Disorders of choroid and retina | 147 (1.0%) |
| Disorders of optic nerve and visual pathways | 98 (0.6%) |
| Effects of foreign body entering through natural orifice | 80 (0.5%) |

monthly presentations ranged from 200 to 400, relatively consistent with the findings from other international studies [9, 10].

The mean age in our study was relatively lower (25.5) than the ones reported in other studies (51.5 [2], 40 [5], and 28.2 [8]). Our study revealed that adult patients comprise only 49% of our study population, in contrast to previous studies, which reported higher numbers for adults, such as Cheung et al. study (89%) [1], Alabbasi et al. study (77.5%) [7], Nanji et al. study (80.7%) [10], Ribeiro et al. study (88.9% of patients were 16 and above) [13]. The most commonly affected group in our study aged <18 years (51%), which differs from those reported in other studies (11% in Cheung et al. study [1] and 19.3% in Nanji et al. study [10]). This difference might be due to the presence of a specialized center for children care (King Abdullah Specialized Children Hospital, KASCH) with more experienced physicians than in any other hospital in Riyadh, potentially making a significant increase in the portion of young patients in our study compared to the other studies mentioned previously.

Regarding the distribution of most common presentations, conjunctivitis weighted for the highest percentage (29.8%) of presentations with a number of 4,559 patients, and was the most common diagnosis, similar to the findings of several studies [11], which showed conjunctivitis to be the most common presentation, including Sridhar et al. (8.7%) [4], Salti et al. (33.75%. in 1997 and 31.75% in 2012 [6], Alabbasi et al. (32.5%) [7], Mahjoub et al. (7.9%) [12], Ribeiro et al. (34%) [13], and de souza Carvalho and José (29.4%) [14]. This can be explained as conjunctivitis has many causes including infections and allergies, which can be a seasonal disease affecting many people [15]. Corneal abrasion, came with 3,570 presentations, held the second most frequent diagnosis (23.3%), comparable to Alotaibi et al. (19.7%) [8] and Jones et al. [16]. However, most other studies reported lower rates of corneal abrasion compared to our study, for example, Channa et al. (13.7%) [3], Sridhar et al. (6.6%) [4], Alabbasi et al. (9.3%) [7], Nanji et al. (12.2%) [10], and Mahjoub et al. (5.6%) [12]. The reason behind this increase might be due to the large number and composition of young patient group (<18 years) (51%) in our study as explained earlier. This age group is physically active at home and school and participate in various sport activities, potentially resulting in trauma including corneal abrasion [17]. Unspecified disorder of eye and adnexa was the third most prevalent diagnosis, with a percentage (12%) a bit higher than that reported in Nanji et al. study that was 9.2% [10].

Diagnoses like unspecified disorder of eye and adnexa restrict the ability for comparison with other studies, as most of the previous studies have not used the ICD-10 codes. Future studies should use a common classification system, such as ICD-10, to make results more accurate and more precise to compare their findings with results from other studies.

### Limitations, strengths, an implications

This study has some evident limitations that need to be highlighted. Firstly, the study obtained and examined data from only a single center. This might limit the generalizability of the findings, as the results may be influenced by specific local practices, patient demographics, and emergency department response to the diseases. Consequently, the outcomes observed may not be applicable to other settings or populations. Additionally, the retrospective nature of the study introduces several inherent limitations. Retrospective studies rely on previously recorded data, which can be incomplete, inconsistent, or biased. There is also the potential for selection bias, as the data available is limited to what was documented and stored, potentially overlooking relevant variables that were not captured at the time. Together, these limitations suggest that while the study provides valuable insights, further research involving multiple centers and prospective designs would be necessary to validate and broaden the applicability of the findings. One of the strengths of this research is the large sample size, with over 15,000 patients examined. This extensive data set enhances the statistical power of the study, allowing for more reliable conclusions. Additionally, the large sample size increases the likelihood that the findings are representative of the broader population, providing greater confidence in the generalizability of the results. Conducting this study in Saudi Arabia adds a significant regional value, as it is the first of its kind in the country and likely in the Middle East. The findings of this study reported the most common population presenting with ocular-related emergencies and what type of emergencies they usually present with when visiting the emergency department in Saudi Arabia. This could add a great value to policymakers and healthcare workers in emergency care settings to understand what type of patients with ocular emergencies they will encounter in clinical practice and how to improve resource allocation and clinical practice to better serve this population.

### Conclusion

As Emergency departments are consistently bustling with activity, it is crucial to improve the understanding and prompt identification of ocular-related conditions, as they hold significant importance due to the potential risks they pose to vision and overall eye health, and to provide appropriate management for each condition. The findings of this study showed a higher incidence of emergencies in patients aged <18 years, with conjunctivitis and corneal abrasion being the most frequently diagnosed conditions for all patients. These findings emphasize the importance of healthcare professionals working in emergency care settings with the required knowledge and skills necessary to effectively manage these common ocular emergencies and improve outcomes. Further larger-scale study is needed to better understand the presentation of ocular-related emergencies in Saudi Arabia.

### Author Contributions

**Conceptualization:** Abdulmohsen Alshammari, Abdulaziz Gharawi, Fahad Almalik, Abdullah Alshibani.

**Data curation:** Abdulmohsen Alshammari, Naif Alqadhy, Abdulaziz Gharawi, Bader Alqahtani, Saif Alagha, Mohammed AlShenaiber, Fahad Almalik, Abdullah Alshibani.

**Formal analysis:** Abdulmohsen Alshammari, Naif Alqadhy, Abdulaziz Gharawi, Bader Alqahtani, Saif Alagha, Mohammed AlShenaiber, Fahad Almalik, Abdullah Alshibani.

**Investigation:** Abdullah Alshibani.

**Methodology:** Abdulmohsen Alshammari, Abdulaziz Gharawi, Bader Alqahtani, Saif Alagha, Fahad Almalik, Abdullah Alshibani.

**Supervision:** Abdullah Alshibani.

**Writing – original draft:** Abdulmohsen Alshammari, Naif Alqadhy, Bader Alqahtani, Saif Alagha, Mohammed AlShenaiber, Abdullah Alshibani.

**Writing – review & editing:** Abdullah Alshibani.

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
