## [Decision Letter · Decision Letter 0]

1 Jul 2024

PONE-D-24-22246Characteristics of Ocular-related Emergency Department Visits: Five-Years Data from a Tertiary Care Center in Riyadh, Saudi ArabiaPLOS ONE

Dear Dr. Alshibani,

Thank you for submitting your manuscript to PLOS ONE. After careful consideration, we feel that it has merit but does not fully meet PLOS ONE’s publication criteria as it currently stands. Therefore, we invite you to submit a revised version of the manuscript that addresses the points raised during the review process.

**ACADEMIC EDITOR: Please provide additional information as requested by reviewer 2.**==============================

We look forward to receiving your revised manuscript.

Kind regards,

Vaitheeswaran G Lalgudi, M.D.

Academic Editor

PLOS ONE

Journal Requirements:

2. For studies involving third-party data, we encourage authors to share any data specific to their analyses that they can legally distribute. PLOS recognizes, however, that authors may be using third-party data they do not have the rights to share. When third-party data cannot be publicly shared, authors must provide all information necessary for interested researchers to apply to gain access to the data. (https://journals.plos.org/plosone/s/data-availability#loc-acceptable-data-access-restrictions) 

Additional Editor Comments:

Please address the comment from reviewer 2 requesting for additional information

Reviewers' comments:

Reviewer's Responses to Questions

**Comments to the Author**

1. Is the manuscript technically sound, and do the data support the conclusions?

Reviewer #1: Yes

Reviewer #2: Yes

2. Has the statistical analysis been performed appropriately and rigorously? 

Reviewer #1: Yes

Reviewer #2: Yes

3. Have the authors made all data underlying the findings in their manuscript fully available?

Reviewer #1: Yes

Reviewer #2: Yes

4. Is the manuscript presented in an intelligible fashion and written in standard English?

Reviewer #1: Yes

Reviewer #2: Yes

5. Review Comments to the Author

Reviewer #1: Well written retrospective epidemiological study relevant to the local region of Saudi Arabia.

Reviewer #2: The study is comprehensive and well done

Entry and exit criteria and how to collect samples are well explained.

The text tables well represent the statistics obtained from the analyses.

In the discussion section, the cases have been well compared and analyzed with other studies.

As a suggestion, it is better to give more details in cases with significant statistics that are the same or different from the rest

6. PLOS authors have the option to publish the peer review history of their article (what does this mean?). If published, this will include your full peer review and any attached files.

Reviewer #1: No

Reviewer #2: No

---

## [Author Response · Author response to Decision Letter 0]

15 Aug 2024

Dear Editor,

On behalf of the authors, I would like to thank you for considering our manuscript for publication in the PLOS ONE journal. We would also like to thank all reviewers for offering their valuable time and effort to review our paper. The reviewers’ comments and feedback are very helpful, and we hope our responses will have improved the paper.

Reviewers’ Comments:

Reviewer 1 comments: Well written retrospective epidemiological study relevant to the local region of Saudi Arabia.

Our response: Thank you for your insightful feedback and positive comments.

Reviewer 2 comments: The study is comprehensive and well done

Entry and exit criteria and how to collect samples are well explained.

The text tables well represent the statistics obtained from the analyses.

In the discussion section, the cases have been well compared and analyzed with other studies.

As a suggestion, it is better to give more details in cases with significant statistics that are the same or different from the rest.

Our response: We appreciate your positive comments and feedback about our study. We agree with your suggestion, and we added the statistics of the studies in the discussion section that either the same or different from the rest. Please revise the manuscript file. 

Thank you all again for your comments and feedback. We hope that we have addressed all comments to improve our paper.

If you have any other questions, please do not hesitate to ask.

Yours sincerely,

Dr. Abdullah Alshibani 

Assistant professor in Emergency Medical Services

---

## [Editor Report · Decision Letter 1]

27 Aug 2024

Characteristics of Ocular-related Emergency Department Visits: Five-Years Data from a Tertiary Care Center in Riyadh, Saudi Arabia

PONE-D-24-22246R1

Dear Dr. AlShibani,

We’re pleased to inform you that your manuscript has been judged scientifically suitable for publication and will be formally accepted for publication once it meets all outstanding technical requirements.

Kind regards,

Vaitheeswaran G Lalgudi, M.D.

Academic Editor

PLOS ONE

---

## [Editor Report · Acceptance letter]

9 Sep 2024

PONE-D-24-22246R1 

PLOS ONE

Dear Dr. Alshibani, 

I'm pleased to inform you that your manuscript has been deemed suitable for publication in PLOS ONE. Congratulations! Your manuscript is now being handed over to our production team.

Kind regards, 

on behalf of

Dr. Vaitheeswaran Ganesan Lalgudi 

Academic Editor

PLOS ONE